# Educational Trials to Quantify Agronomic Information in Interdisciplinary Fieldwork in Pursat Province, Cambodia

**Rongling Ye** [1], **Taisuke Kodo** [2], **Yoshihiro Hirooka** [3], **Hor Sanara** [4], **Kim Soben** [5], **Satoru Kobayashi** [6] **and Koki Homma** [1,2,*]

1 Graduate School of Agricultural Science, Tohoku University, Sendai 980-8572, Japan
2 Graduate School of Agriculture, Kyoto University, Kyoto 656-8502, Japan
3 Graduate School of Agriculture, Kindai University, Nara 631-8505, Japan
4 Faculty of Land Management and Land Administration, Royal University of Agriculture, Phnom Penh 120501, Cambodia
5 Faculty of Forest Science, Royal University of Agriculture, Phnom Penh 120501, Cambodia
6 Centre for Southeast Asian Studies, Kyoto University, Kyoto 606-8501, Japan
* Correspondence: koki.homma.d6@tohoku.ac.jp; Tel.: +81-22-757-4083; Fax: +81-22-757-4087

**Abstract:** Improving agricultural research and education is highly recommended to control agricultural development and environmental sustainability in Cambodia. Agricultural research mostly focuses on interviews with farmers as a first measure in developing countries, but a lack of quantitative accuracy remains one of the major constraints. In this situation, we conducted educational activities for master's degree students of the Royal University of Agriculture (RUA) to append agronomic information with popular equipment in interdisciplinary fieldwork in Pursat Province, Cambodia. For the popular equipment, an RGB camera, a reflectometer as well as pH and EC meters were selected. The agronomic information collected by the students supported the results obtained during the interviews. For example, the difference in fertilizer application between the irrigated and nonirrigated areas was confirmed by the soil ammonium concentration evaluated with a reflectometer; the difference in rice growth among water conditions was confirmed by the leaf area percentage evaluated with an RGB camera. Since the majority of the students lacked agricultural and statistical knowledge, the agronomic information quantified by popular equipment provided proper educational materials. The interdisciplinary fieldwork also indicated serious problems in the study area, such as low beneficial crop production and environmental sustainability. To overcome these problems, improving agricultural education is required to foster skillful agricultural professionals, and the quantification of agronomic information is an essential issue.

**Keywords:** Cambodia; educational trail; farmland; crop growth; soil characteristics

## 1. Introduction

Cambodia is one of the countries that has achieved rapid economic development [1], but it is still an agricultural country with a population of approximately 14 million, 77% of which lives in rural areas [2]. Economic development also stimulated agricultural production to attain a great increase, but 16.4% of the total population is still undernourished [2]. Since only 2.8 of the 6.7 million ha of arable land is cultivated, Cambodia still has great potential for agricultural development. However, disordered agricultural development causes severe deforestation, which reduces sustainability in the area [3]. Balance between agricultural development and environmental conservation is of crucial importance. This unbalanced development is partly due to limited numbers of skillful agricultural professionals, such as researchers and extension workers. Improving the agricultural research and education level is important to fundamentally address this problem.

The development of scientific research on agriculture is a significant challenge in Cambodia today. Higher education started developing in the 1990s after the disruption caused

by the totalitarian revolution, warfare, and international isolation in the 1970s and 1980s. As Un and Sok [4] indicated, higher education investment from Cambodia's government is much lower than the world average. Many university faculties lack qualifications and opportunities to improve their skills; thus, teaching depends on rote learning without an examination of empirical evidence [4,5]. The government considers agriculture as the axis of economic development, but the agriculture academia is limited in number [6] and the agricultural extension system is underdeveloped. Cambodian rural society is the same as other countries in the globalizing world. Cambodian farmers started to use modern technologies and goods for agricultural production in the 2000s. However, they make decisions based on the information from sellers or the narratives of other farmers who have no technical qualification [7]. This condition is problematic in conceiving erroneous farming practices [8]. For example, many farmers tend to overuse pesticides and seeds for a higher yield, which can only increase the cost and environmental pollution without increasing economic income [7]. The only solution to this problem is to educate adequate personnel for agricultural extension workers who understand the unique characteristics of each production environment through scientific knowledge in agriculture.

To contribute to the improvement in agricultural research and education, the authors held workshops entitled "Interdisciplinary Fieldwork for Sustainable Livelihoods Studies" in Pursat Province, Cambodia, under a joint educational program between the Royal University of Agriculture (RUA), Cambodia, and the Center for Southeast Asian Studies (CSEAS), Kyoto University, Japan. The workshops aimed to evaluate livelihood transition under rapid socioeconomic change in rural Cambodia. To this end, several rounds of fieldwork were conducted with graduate school students in RUA. Pursat is located in the western part of Cambodia and has a diverse biophysical environment, including the inundated area of Tonle Sap Lake and lowland and mountainous areas. Accordingly, the fieldwork covered several aspects, such as household economy, migration, fishery, land management, forest management and crop production. Originating from the activities, several studies have been published [9–14].

This kind of interdisciplinary fieldwork has mostly been organized through interviews in developing countries. Since systematic information gathering, such as national statistics, is generally inadequate in developing countries, interviews are thus a primary and casual measure. However, the information based on interviews lacks quantitative accuracy, which is critical, especially in relation to agricultural production. Under the above situation, we conducted educational activities to append agronomic information with popular equipment accompanying the interviews. This report explained the methodology and discussed its educational effects.

## 2. Materials and Methods

### 2.1. Outlines of Education Activity

The activities were conducted in Pursat, Cambodia, from 18 to 24 August 2014, 7 to 13 September 2015 and 25 to 30 September 2017. The targets of activities changed each period: the difference between irrigated areas and adjacent nonirrigated areas was focused on in 2014; the relation between water conditions and crop management was focused on in 2015; whilst crop management in a mountainous area was focused on in 2017. A total of 17, 12 and 10 students on master's courses at Natural Resource Management, Royal University of Agriculture, Cambodia, joined the activities, respectively. Students were grouped into several groups to investigate the fields and to interview farmers under our instructions. The interview items commonly included cultivation management, such as fertilizer use and rice cultivars.

The investigation in 2014 was conducted in an area irrigated by the Krauch-Sauch channel and its adjacent nonirrigated area located in the Khnar Totueng commune and Trapeang Chorng commune, Bakan district. Seventeen farmers were selected from both the irrigated area and nonirrigated area. In 2015, four types of communes were selected for the investigation: Svay Doun Kaev commune in the flood-prone area; Boeng Khnar

commune in the irrigated rice area; Rumlech commune in the rainfed rice area located in Bakan district; and Talo commune in a drought-prone area located in Talo Sen Chey district. The investigation was conducted with four farmers from each commune. In 2017, the investigation was conducted for 18 farmers in three villages, namely Dey Kraham, Chamkar Chrey Khang Tbaung (Chamker Chrey) and Ou17 in Anlong Reap commune, Veal Veng district.

## 2.2. Student Activities to Collect Agronomic Information

### 2.2.1. Crop Growth

A photograph taken from the top of the canopy is typically utilized as an economical method to assess the leaf coverage of crops [15,16], whilst that taken from the bottom of the canopy is utilized to assess the leaf area of trees [17,18]. We instructed the students on the method to take photographs vertically downward from the top and vertically upward from the bottom of the canopy with a monopod (Figure 1). For this research, digital cameras (Coolpix W100, Nikon Co., Tokyo, Japan) were used. The subject crops in the study were rice in 2014; rice and cassava in 2015; and upland rice, maize, sugarcane, cassava and soybean in 2017. The numbers and kinds of crops varied depending on the interviewed farmers. Three photographs each from the top and from the bottom were taken for each subject crop.

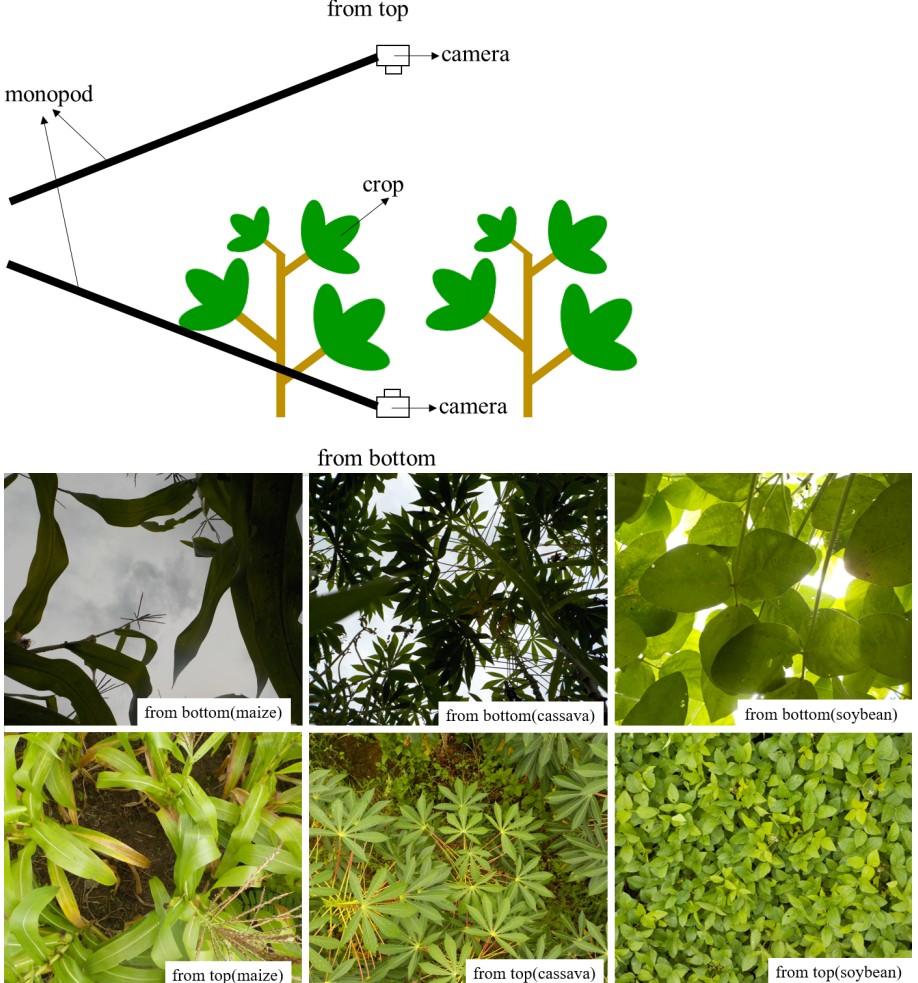

**Figure 1.** Canopy photographs taken by a digital camera downward from the top and upward from the bottom of the canopy with a monopod.

We instructed students to analyze photographs using the free image analysis software (ImageJ, version 1.53a, NIH, Bethesda, MD, USA; [19]) at every investigation in 2014, 2015 and 2017. Figure 2 shows the process: the image was split into three R, G and B channels, and converted to the grayscale image by the following formula $(G - B) + (G - R)$, divided into leaves and others by manually setting a threshold, and then the leaf area percentage was calculated [20].

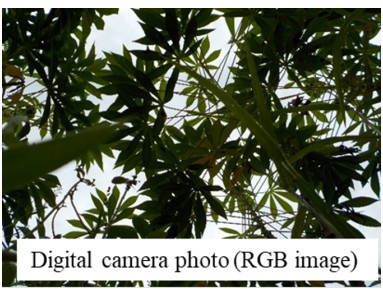

Digital camera photo (RGB image)

↓split channels

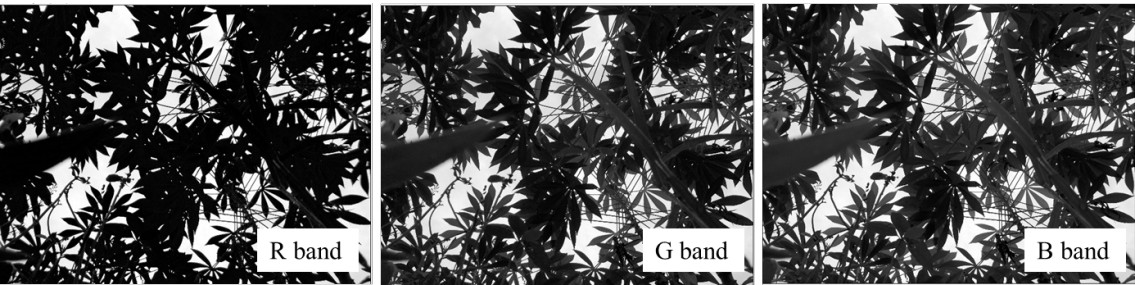

R band    G band    B band

↓grayscale image by formula $(G - R) + (G - B)$

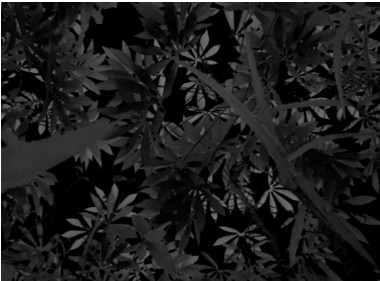

↓divide into leaf and the others

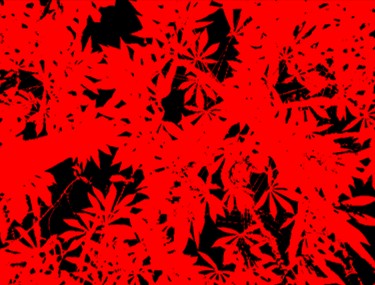

**Figure 2.** Process of calculating leaf coverage from photographs.

2.2.2. Soil Characteristics

Soil samples were taken from the surface to 15 cm below the ground using a shovel. Students characterized the soil type by observation following the protocol under our in-

struction [21]. In 2014, soil samples were taken from each field. After diluting with distilled water and filtering, water-extracted soil ammonium was measured with a reflectometer (RQflex10 plus, Merck Co., Kenilworth, NJ, USA) [22]. In 2015 and 2017, an image analysis method was instructed to the students to evaluate the soil color. The soil color was evaluated based on the CIELAB system, which expresses the color as 3 coordinates, L*, a* and b*: L* denotes the lightness from black (0) to white (100); a* is the coordinate from green (−) to red (+); and b* is the coordinate from blue (−) to yellow (+) [23]. Small volumes of each soil sample were placed on a white paper background for photographing. L*a*b* values were evaluated for the photographs by ImageJ. In 2017, the electrical conductivity (EC) and pH of a 1:5 soil–water extract with bottled drinking water (EC 26 mS/cm, pH 6.4) were measured using a portable EC meter (LAQUAtwin-EC-33B, HORIBA Ltd., Kyoto, Japan) and pH meter (LAQUAtwin-pH-11B, HORIBA Ltd., Kyoto, Japan).

### 2.3. Supplemental Measurement

We measured the leaf area index (LAI: leaf area per unit land area) and normalized difference vegetation index (NDVI; [24]) to verify the leaf area percentage calculated from the photographs. LAI was measured by a canopy analyzer (LAI-2200C, LI-COR Inc., Lincoln, NE, USA) in 2014 and 2017. NDVI was measured by a spectral meter (MS-720, EKO Instruments Co. Ltd., Tokyo, Japan) in 2017.

## 3. Results

### 3.1. Verification of Leaf Area Percentage

The leaf area percentage calculated from the photographs by the students was compared with LAI in 2014 and LAI and NDVI in 2017 measured by the authors (Figure 3). Figure 3a shows that the LAI of rice was more closely related to the leaf area percentage from the bottom. The leaf area percentage from the top leached a higher percentage at a lower LAI than that from the bottom. Figure 3b,c verified that the leaf area percentage from the bottom was related to LAI and from the top was related to NDVI. The leaf area percentage of upland rice was lower than that of the other crops with the same LAI. The relationships between the leaf area percentage and NDVI showed little difference among crops.

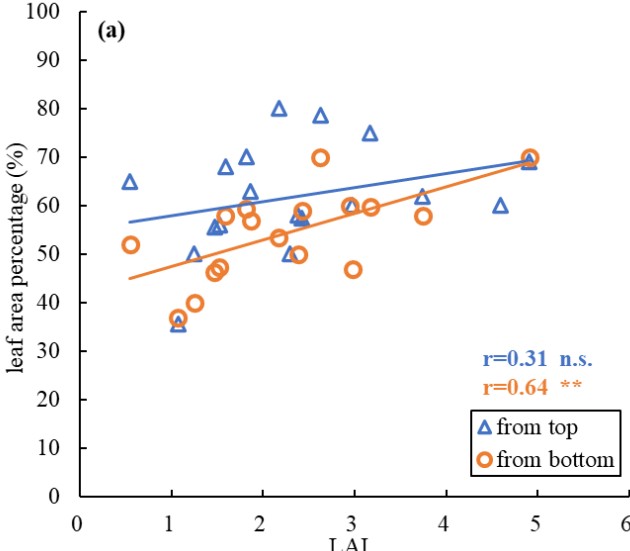

**Figure 3.** *Cont.*

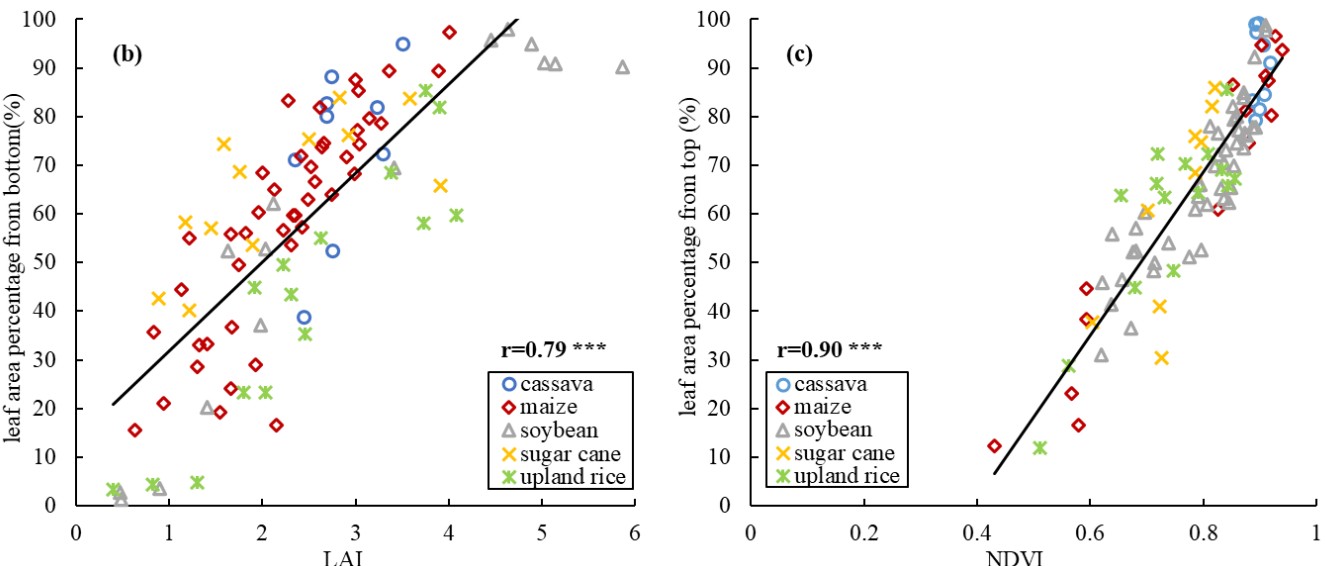

**Figure 3.** (**a**) Linear relationship between the leaf area percentage and LAI of rice in 2014; and (**b**) Linear relationship between the leaf area percentage from bottom and LAI in 2017; and (**c**) Linear relationship between leaf area percentage from top and NDVI in 2017. ***: *p*-value < 0.01, **: 0.01 < *p*-value < 0.05, n.s.: no significant difference.

### 3.2. Summary of Interviews

The interviews in 2014 revealed that the farmers in irrigated areas applied more fertilizer than those in nonirrigated areas [10]. Irrigation provides the availability of the earlier planting of aromatic rice for sale. The farmers in the nonirrigated area mainly produced rice for self-sufficiency.

The difference between farmers in irrigated rice areas and rainfed rice areas was also observed in 2015. Farmers in flood-prone areas had constraints of deep water for rice production. They traditionally planted rice in the rainy season, but they started to plant rice in the dry season [14]. The popularization of gasoline engine pumps enhanced the change from rainy season rice to dry season rice. Some rice in drought-prone areas showed symptoms of drought.

The different main cultivated crops were revealed in the 2017 investigation: maize was the main crop in Dey Kraham, soybean was the main crop in Ou17, and the maize and rice were the main crops in Chamkar Chrey. Farmers in Chamkar Chrey had already used chemical fertilizer, but farmers in Dey Kraham and Ou17 had not yet used chemical fertilizer. Most farmers decided the sowing date according to their own judgment, resulting in completely different sowing dates among farmers of the same crop.

### 3.3. Agronomic Evaluation by Student Activities

Soil analysis revealed that the soil ammonium concentration was different between the irrigated area and nonirrigated area in 2014 (Figure 4). However, the difference in leaf area percentage was small (Figure 5).

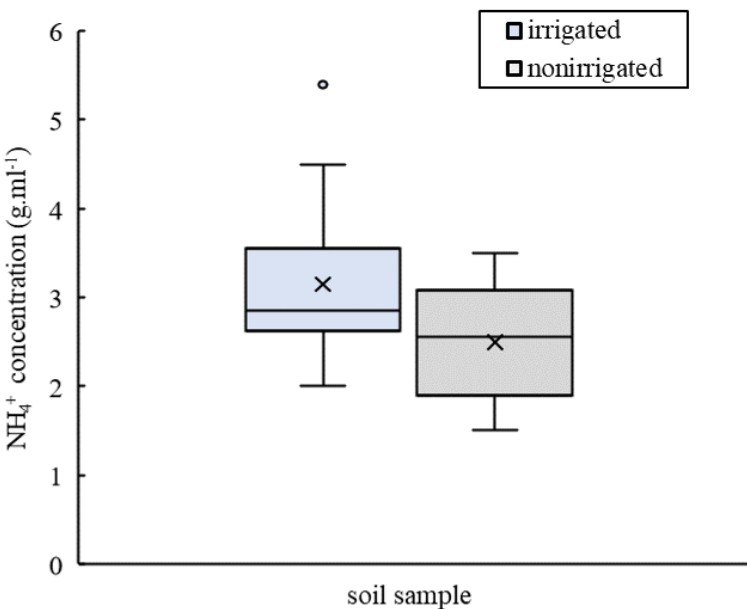

**Figure 4.** Box plot of ammonium nitrogen concentration in soil samples of irrigated area and nonirrigated area in 2014. Circle indicates outlier.

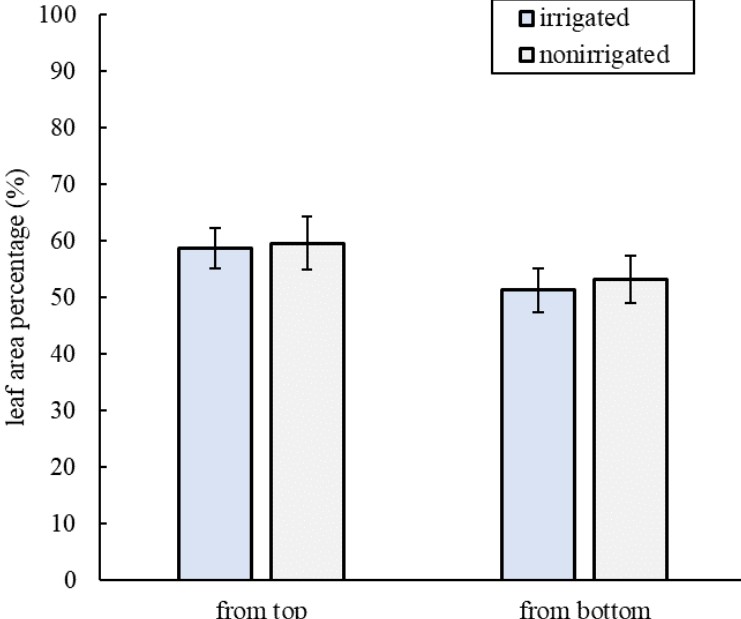

**Figure 5.** Leaf area percentage of rice in irrigated areas and nonirrigated areas in 2014. Error bars indicate s.e.

In 2015, the largest leaf area percentage was observed in the irrigated rice area, followed by the rainfed rice area and drought-prone area (Figure 6). The smallest leaf area percentage was observed in the flood-prone area. Soil color formed a cluster of each commune (Figure 7).

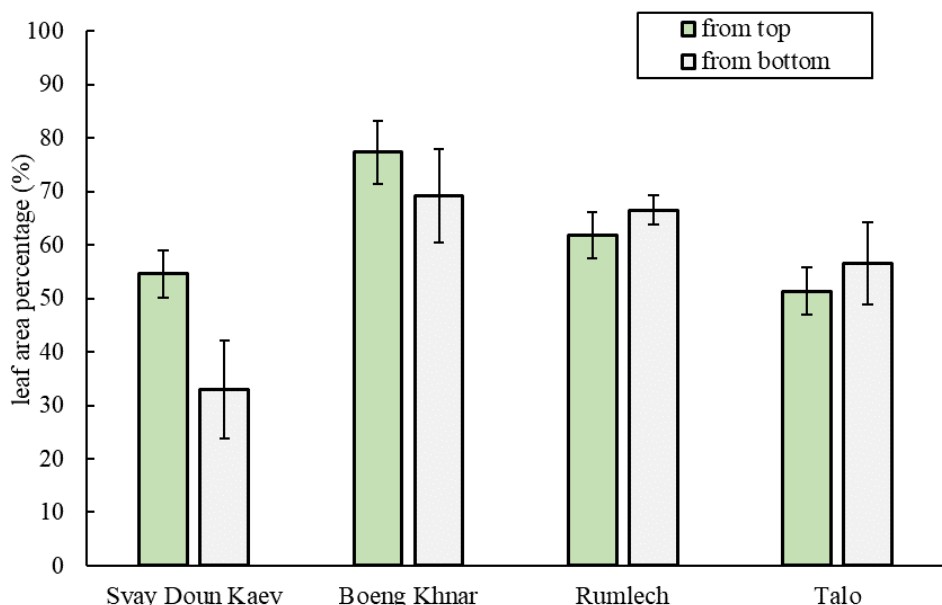

**Figure 6.** Leaf area percentage of rice in different areas in 2015. Svay Doun Kaev, Boeung Khnar, Rumlech and Talo communes are located in flood-prone, irrigated rice, rainfed rice and drought-prone areas, respectively. Error bars indicate s.e.

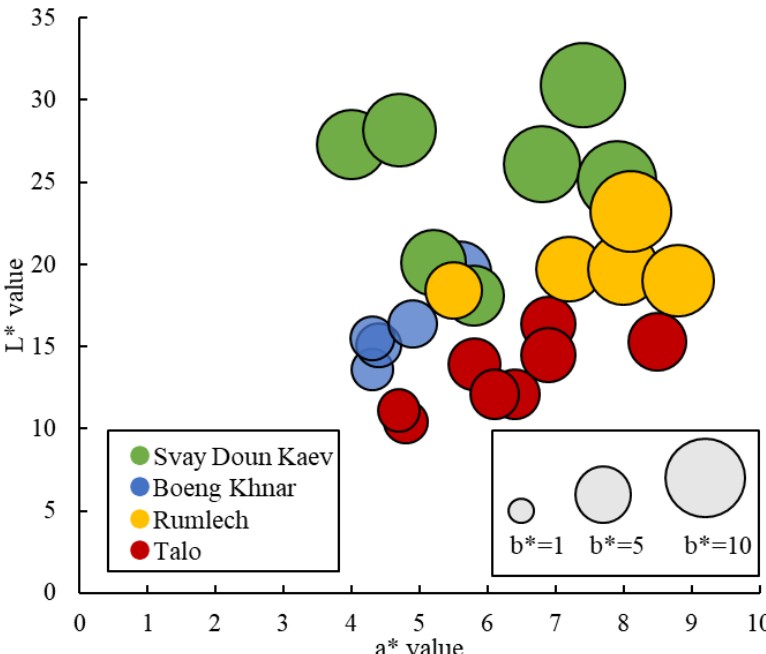

**Figure 7.** L*a*b* values of soil color in different communes in 2015. The area of bubbles indicates b* values.

A wide range of leaf area percentages of the same crop were observed in 2017, among which soybean had the largest range, from 12.4% to 96.5%, followed by upland rice and maize (Figure 8). Although many factors impacted crop growth, fertilized farmland tended to have higher leaf area percentages (Figure 9). Soil color did not form any distinguishable cluster of commune (Figure 10).

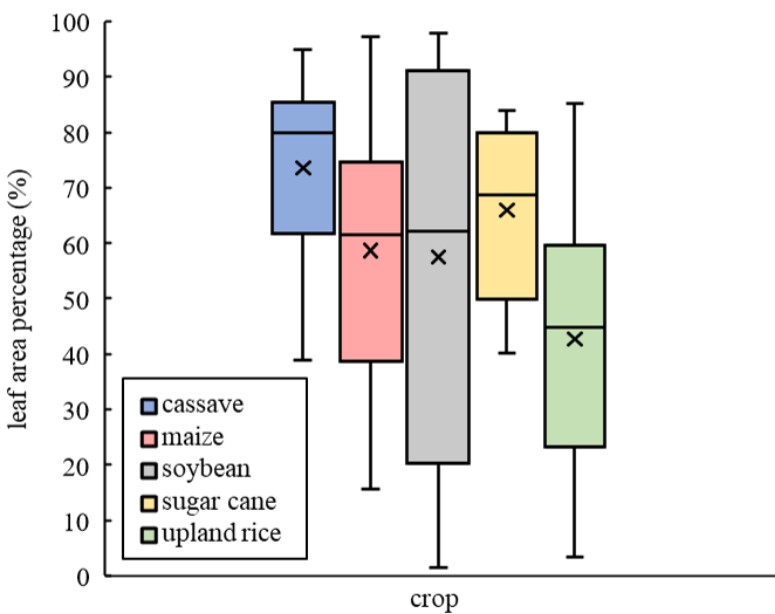

**Figure 8.** Box plot of leaf area percentage from the bottom of crops in 2017.

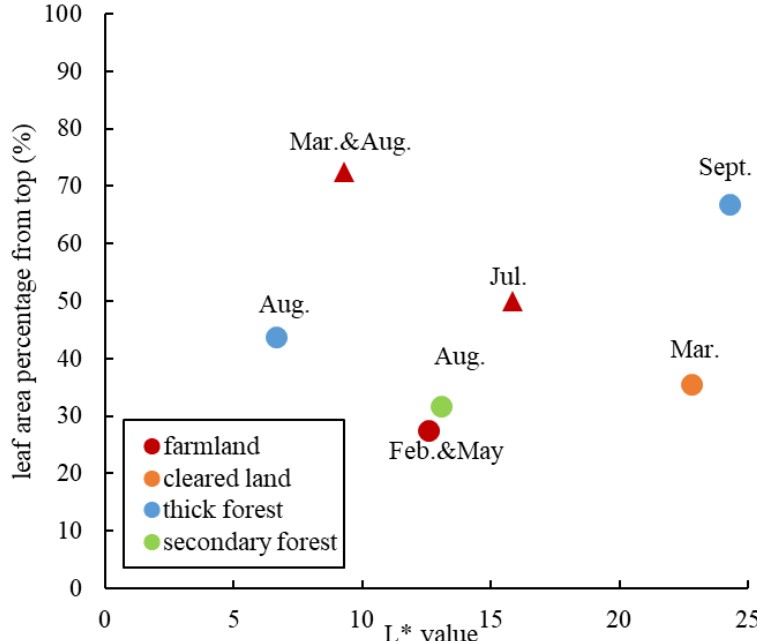

**Figure 9.** Relationship between the leaf area percentage and L* value of soil color in maize field in 2017. Triangles and circles indicate fields with and without chemical fertilizer application, respectively. Color indicates the previous field type during the interview. The sowing month obtained by interviews is shown in the figure.

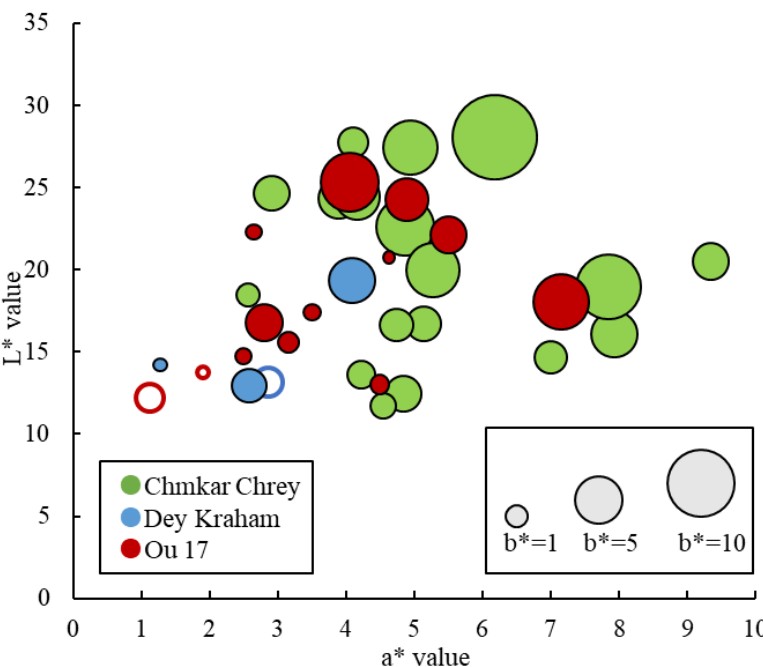

**Figure 10.** L*a*b* values of soil color in different communes in 2017. The area of bubbles indicates b* values. Hollow circles are negative b* values.

## 4. Discussion

The agronomic information collected by the students supported the results obtained from the interviews. The difference in fertilizer application between irrigated and nonirrigated areas reflected the soil ammonium concentration in 2014. The difference in rice growth among communes was quantified in terms of leaf area percentage in 2015, confirming that the water condition was one of the major determining factors in rice production in the study area. The difference in water conditions seems to be reflected in the soil color. The lighter color with a larger L* value might be produced by gleization under water [25]. The leaf area percentage suggested a difference in growth among crops in 2017.

Ordinarily, these kinds of agronomic parameters are measured by expensive instruments [12,26]. The canopy analyzer and the spectral meter cost approximately USD 25,000 and USD 10,000, respectively. Soil nitrogen is often measured with an NC analyzer, which costs approximately USD 50,000. In contrast, the digital cameras, EC meter and pH meter in this study cost approximately USD 200, USD 250 and USD 180, respectively. The reflectometer (RQflex10 plus) was rather expensive, but its cost was USD 1000. Smartphones with digital cameras have also been used as common equipment in developing countries, and the students tested their own in this study. Tanaka et al. developed an application to estimate the rice yield from RGB images at harvest [27]. Such a technological development enhances the utilities of smartphones. Although the accuracy of measurement with popular equipment is inferior to that with expensive equipment, the measurement activity is the highest priority.

Since the availability of free satellite data has been increasing, many regional studies have used them to evaluate agriculture in developing countries [13,14,28,29]. Some students who joined in this study had an education on satellite data and knew the NDVI. However, they did not exactly know the relation between NDVI and LAI. The leaf area percentage varies depending on whether it is evaluated from the bottom or the top of the canopy. Theoretically, the leaf area percentage from the top is larger than that from the bottom because the leaves develop to receive more sunlight. That from the top more closely corresponds to the NDVI and that from the bottom closer corresponds to the LAI. The relation is partly shown in Figure 3. The nontheoretical relation, such as the larger leaf area percentage from the bottom than from the top, was partly derived from the quality of the

photograph. Some photographs taken from the bottom suffered from overexposure, and some from the top had difficulty in "divide into leaf and the others" process. We should carefully instruct the students to check the quality of the photographs. We realized that the common lines of reasoning in scientific research also need to be taught, such as confirming the data once investigated.

Soil color is an important factor in soil research and is relatively easy to evaluate [21]. We emphasized the necessity of collecting soil color data using the CIELAB system for students. Students recorded soil color by subjective judgment, such as black and red, leading to incomplete information. L*a*b* values enabled students to quantify soil colors to more intuitively distinguish different soil characteristics.

In the students' feedback, they were interested in joining the field investigation and collecting agronomic information, as they had few chances of talking with the farmers and directly observing the farmland. Compared with a simple social survey, adding agronomic information enables them to better understand the interview results in relation to crop production. The majority of the students felt more confident and enthusiastic about future research. Although all students showed high enthusiasm for learning during all activities, their summary reports had many errors due to the lack of agricultural and statistical knowledge. The inconsistency between the sowing date and leaf area percentage in Figure 9 partly resulted from insufficient agricultural knowledge. They should confirm the sowing month to the farmers if their answer and crop condition did not match. These matters also emphasize that education in actual farmer fields is quite important and that the quantification of agronomic information should be even conducted by utilizing popular equipment.

The interviews suggested many problems in the study area. First, most farmers regarded the market as the only basis in deciding the main crop, regardless of whether the production environment was suitable. Second, the majority of farmers used credits for crop production, which easily turned into debt because the market and production were both unstable. Third, despite introducing agricultural machinery and chemicals, cultivation management tended to be extensive due to labor shortages by migration. Fourth, environmentally friendly management, such as biochar and green manure application [30–33], was not popular. These problems appeared to reduce the sustainability of crop production from a social and environmental perspective. To overcome this situation, fostering agricultural extension workers and researchers a first recommended. Training that teaches common sense in scientific research, basic knowledge of agriculture, and analysis should be initiated for future developments. Each technique proposed in this study is well known but rarely implemented. Accordingly, training should incorporate the importance of quantifying agronomic information by utilizing popular equipment. International and interdisciplinary collaborations such as this project can contribute to improving the educational situation in Cambodia.

**Author Contributions:** Conceptualization, K.H.; methodology, K.H.; software, R.Y.; formal analysis, R.Y.; investigation, K.H., R.Y., T.K., Y.H., H.S., K.S. and S.K.; data curation, R.Y.; writing—original draft preparation, R.Y.; writing—review and editing, K.H.; visualization, R.Y.; supervision, K.H. All authors have read and agreed to the published version of the manuscript.

**Funding:** This work was partly supported by JSPS KAKENHI, Grant Numbers 15H05144, 19H00559, 19H03069, JST SPRING, Grant Number JPMJSP2114, and the educational program was supported by the CSEAS of Kyoto University, Japan.

**Institutional Review Board Statement:** Not applicable.

**Informed Consent Statement:** Not applicable.

**Data Availability Statement:** Not applicable.

**Acknowledgments:** We are grateful to the students of the Royal University of Agriculture for data and interview. We are also grateful to the local farmers for their cooperation.

**Conflicts of Interest:** The authors declare no conflict of interest.

**Abbreviations**

The following abbreviations are used in this manuscript:

| | |
|---|---|
| NDVI | Normalized difference vegetation index |
| LAI | Leaf area index |
| EC | Electrical conductivity |

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
