# Peer review of "Educational Trials to Quantify Agronomic Information in Interdisciplinary Fieldwork in Pursat Province, Cambodia"

_sustainability, doi:10.3390/su141610007_

Round 1

Reviewer 1 Report

This manuscript needs to be improved so that it can be considered for publication.

The introduction needs to be extended with the most important knowledge in this area.

Line 84: Enter the dates correctly.

Line 90: Describe more detailed instrument settings as well as measuring technique.

Line 92: Only small typo. The space between the word and the citation.

Figure 7: It should be made better so that the differences are clear.

Line 181, fertilizer application: This issue has been addressed in great detail by this very important paper and therefore the authors are encouraged to add it here as a reference. https://www.sciencedirect.com/science/article/pii/S0048969722011354

Ensure the correct reference format.

Author Response

This manuscript needs to be improved so that it can be considered for publication.

Thank you for your comments. We revised by following your comments.

The introduction needs to be extended with the most important knowledge in this area.

We added one paragraph about educational situation in Cambodia into introduction (Line55-72).

Line 84: Enter the dates correctly.

We fixed the way to write date (Line96).

Line 90: Describe more detailed instrument settings as well as measuring technique.

We added the instrument setting and measuring technique (Line147).

Line 92: Only small typo. The space between the word and the citation.

Revised.

Figure 7: It should be made better so that the differences are clear.

We tried to improve Figure 7. Black line was added to circles. Relation between size of circle and value of b* was added.

Line 181, fertilizer application: This issue has been addressed in great detail by this very important paper and therefore the authors are encouraged to add it here as a reference. https://www.sciencedirect.com/science/article/pii/S0048969722011354

We mentioned the importance of environmentally friendly management and add the proposed reference (Line303).

Ensure the correct reference format.

We corrected reference format.

Reviewer 2 Report

- English writting must be improved

- Discussion must be improved (use new related articles)

Author Response

- English writing must be improved

We fully revised the manuscript and had another English proofreading service.

- Discussion must be improved (use new related articles)

We emphasized our achievement in the current education in Cambodia (L295-297, L309-314). We also added current situation of agricultural education in the introduction (L55-73). Several recent articles were added to the reference (no. 3-8 and 33-33).

Reviewer 3 Report

This work has high relevance and scientific merit. I believe that the approach applied at work highlights its importance. In general, I recommend that a review be carried out regarding the standardization of formatting (size and font) and description of the legends and nomenclature of the x and y axes of the figures presented. Additionally, I suggest that a grammar and typing review (proof reading) be applied.

Author Response

We appreciate your approval for our manuscript. The figures were fully revised, and another English proofreading service was applied.

Reviewer 4 Report

Sustainability-1801676

YE et al  submitted the manuscript entitled “Educational trials to quantify the agronomic information in interdisciplinary fieldworks in Pursat Province, Cambodia “for publication consideration in Sustainability

Their study aimed to provide knowledge of improvement of agricultural research and education which could improve agricultural productivity and to reduce the undernourished population in developing countries like Cambodia. The conclusion of this report is based on interdisciplinary fieldwork conducted by local agricultural graduate students exhibited some problems. Therefore, improvement of agricultural education is required to foster skillful agricultural professionals, and the quantification of agronomic information is critical.

Authors tried to classify the relationships between professional education including educate rural worker and agronomic information collection/analysis by defining study areas. It is very encouraging idea to help these developing countries to sustain agricultural development. 

However, I found the manuscript’s academic contribution is weak. It should follow principle of “hypothesis-experiment verification-conclusion” and repeatability of study as sound research article. The study meets the aims and scope of Sustainability journal. 

Another issue is the English language is not easy to understand. For example, in line1,2:

substantially recommended to sustainably develop agricultural productivity and to reduce the undernourished population in Cambodia”, not sure what are authors’ points here.

Line 22,23: “Cambodia also started to export several agricultural products with its development, but 16.4% of the total population is still undernourished[2]” . It is not easy to understand what is this sentence’s meaning, Agricultural products? Cambodia’s development? 

There is more example throughout the whole manuscript.

I believe this report is interesting to many, to be accepted for publication, major revision is required including reorganize the major content of manuscript. 

Author Response

YE et al submitted the manuscript entitled “Educational trials to quantify the agronomic information in interdisciplinary fieldworks in Pursat Province, Cambodia “for publication consideration in Sustainability.

Their study aimed to provide knowledge of improvement of agricultural research and education which could improve agricultural productivity and to reduce the undernourished population in developing countries like Cambodia. The conclusion of this report is based on interdisciplinary fieldwork conducted by local agricultural graduate students exhibited some problems. Therefore, improvement of agricultural education is required to foster skillful agricultural professionals, and the quantification of agronomic information is critical.

Authors tried to classify the relationships between professional education including educate rural worker and agronomic information collection/analysis by defining study areas. It is very encouraging idea to help these developing countries to sustain agricultural development.

We are very glad to have your comments. We emphasized our achievement in the current education in Cambodia (L295-297).

However, I found the manuscript’s academic contribution is weak. It should follow principle of “hypothesis-experiment verification-conclusion” and repeatability of study as sound research article. The study meets the aims and scope of Sustainability journal.

We agree the techniques proposed in this study are well known in the academic. However, the actual implementation is rare in the developing countries. We believe our manuscript sill contribute to change such situation. We added sentences to emphasize our proposal (L309-313).

Another issue is the English language is not easy to understand. For example, in line1,2:

“substantially recommended to sustainably develop agricultural productivity and to reduce the undernourished population in Cambodia”, not sure what are authors’ points here.

We revised the sentence:

Improvement of agricultural research and education is substantially recommended to control agricultural development and environmental sustainability in Cambodia. (L17)

Line 22,23: “Cambodia also started to export several agricultural products with its development, but 16.4% of the total population is still undernourished [2]” . It is not easy to understand what is this sentence’s meaning, Agricultural products? Cambodia’s development?

We revised the sentence:

Economic development also stimulated agricultural production to attain a great increase, but 16.4% of the total population is still undernourished [2].

There is more example throughout the whole manuscript.

We added several references (no. 3-8 and 30-33) to support our manuscript.

I believe this report is interesting to many, to be accepted for publication, major revision is required including reorganize the major content of manuscript.

Thank you for your comments. We believe the manuscript was improved based on your comments.

Round 2

Reviewer 1 Report

The manuscript can be accept.

Reviewer 4 Report

The revised manuscript addressed my concerns and questions. It should be suitable for publish in Sustainability.

Best Regards.